# Controlling the coherence of a diamond spin qubit through its strain environment

Young-Ik Sohn [1], Srujan Meesala[1], Benjamin Pingault[2], Haig A. Atikian[1], Jeffrey Holzgrafe[1,2], Mustafa Gündoğan[2], Camille Stavrakas[2], Megan J. Stanley[2], Alp Sipahigil[3], Joonhee Choi[1,3], Mian Zhang[1], Jose L. Pacheco[4], John Abraham[4], Edward Bielejec[4], Mikhail D. Lukin[3], Mete Atatüre[2] & Marko Lončar[1]

The uncontrolled interaction of a quantum system with its environment is detrimental for quantum coherence. For quantum bits in the solid state, decoherence from thermal vibrations of the surrounding lattice can typically only be suppressed by lowering the temperature of operation. Here, we use a nano-electro-mechanical system to mitigate the effect of thermal phonons on a spin qubit – the silicon-vacancy colour centre in diamond – without changing the system temperature. By controlling the strain environment of the colour centre, we tune its electronic levels to probe, control, and eventually suppress the interaction of its spin with the thermal bath. Strain control provides both large tunability of the optical transitions and significantly improved spin coherence. Finally, our findings indicate the possibility to achieve strong coupling between the silicon-vacancy spin and single phonons, which can lead to the realisation of phonon-mediated quantum gates and nonlinear quantum phononics.

---

[1] John A. Paulson School of Engineering and Applied Sciences, Harvard University, 29 Oxford Street, Cambridge, MA 02138, USA. [2] Cavendish Laboratory, University of Cambridge, J. J. Thomson Avenue, Cambridge CB3 0HE, UK. [3] Department of Physics, Harvard University, 17 Oxford Street, Cambridge, MA 02138, USA. [4] Sandia National Laboratories, Albuquerque, NM 87185, USA. These authors contributed equally: Young-Ik Sohn, Srujan Meesala, Benjamin Pingault. Correspondence and requests for materials should be addressed to M.L. (email: loncar@seas.harvard.edu)

Solid state quantum bits can offer the key advantage of scalability when used to realise a quantum network. However, their coherence is often limited by the impact of fluctuations in the solid-state environment. In this context, the effects of fluctuating electric[1–3] and magnetic fields[4–8] on the optical and spin coherence of solid-state emitters can be mitigated by applying a variety of techniques ranging from materials engineering to dynamical decoupling. On the other hand, thermal decoherence can typically be overcome only by maintaining the quantum system at low enough temperatures to freeze out relevant phonons. Phonon-driven processes are responsible for relaxation and decoherence processes in a variety of solid-state emitters that can serve as optically accessible quantum memories[9–15]. In particular, for emitters with spin-orbit coupling, such processes can demand operation at sub-Kelvin temperatures[16–18], or the use of magnetic fields of several Tesla[19] to achieve long spin relaxation and coherence times. This requires cryogenic setups that are significantly more complex than common helium-4 cryostats employed to obtain coherent optical photons from solid-state emitters.

In this work, we show that strain engineering can be used to quench the effect of the thermal phonon bath on an electronic spin qubit without lowering the operating temperature. Our experiments are performed on the negatively charged silicon-vacancy (SiV⁻) centre in diamond, an emerging building block for photonic quantum networks[20] due to its remarkable optical properties stemming from its inversion symmetric structure[21]. However, unless operated at dilution refrigerator temperatures[16,17], the SiV⁻ centre is subject to phonon-induced transitions between its energy levels[11,22] which limit its spin coherence[23]. Our approach to mitigate phonon-induced decoherence takes advantage of the fact that the large electron-phonon coupling responsible for such decoherence processes fundamentally arises from a high susceptibility of the electronic orbitals to lattice strain. Through strain control, we increase the energy scale for phonon absorption by the emitter to far above the thermal energy ($k_B T \approx 0.3$ meV at the experimental temperature, $T = 4$ K). The resulting depletion of thermal phonons seen by the SiV⁻ leads to an improvement in its spin coherence time.

## Results

**Description of the nano-electro-mechanical system.** The detailed level structure of the SiV⁻ centre is depicted in Fig. 1a, with the ground-state (GS) and excited-state (ES) manifolds, each containing two distinct orbital branches[24]. Orbital degeneracy in each manifold is lifted by spin-orbit coupling: $|1\rangle, |2\rangle$ in the GS split by 46 GHz, and $|3\rangle, |4\rangle$ in the ES split by 255 GHz in the absence of strain. Phonons with frequencies corresponding to these splittings can drive orbital transitions within the ground and excited manifolds[11]. As a first step towards controlling the electron-phonon interaction, we investigate the effect of static strain on these orbitals through strain-dependent photoluminescence excitation (PLE) of the optical transitions labelled A, B, C and D at 4 K. Static strain control at the location of the emitter is achieved with a nano-electro-mechanical system (NEMS) device, a monolithic single-crystal diamond cantilever with metal electrodes patterned above and below it (see Supplementary Fig. 1), as shown in the scanning electron microscope (SEM) image in Fig. 1b. An opening in the top electrode allows optical access to SiV⁻ centres located in an array (inset of Fig. 1b), precisely positioned by focused ion-beam (FIB) implantation of $^{28}$Si$^+$ ions[25,26]. A DC voltage applied across the electrodes deflects the cantilever downwards due to electrostatic attraction and generates controllable static strain oriented predominantly along the long axis of the cantilever. The strain profile

can be calculated numerically via a finite-element-method (FEM) simulation, as shown in Fig. 1c. Of the two possible orientations of SiV⁻ centres in our device, we address those with transverse orientation (labelled blue, and shown in detail in inset of Fig. 1c), which predominantly experience strain in the plane normal to their highest symmetry axis ($E_g$-symmetric strain[27]). Upon applying strain, transitions A and D shift towards shorter and longer wavelengths, respectively. These shifts indicate increasing GS and ES splittings as shown in Fig. 2a. This result is consistent with a previous experiment on a dense ensemble of SiV⁻ centres[28]. Complete characterisation of the strain response of the SiV⁻ electronic levels and relevant group theory analysis are detailed in ref[29]. The variations in GS and ES splittings shown in Fig. 2a are quadratic at low strain, and linear at high strain. This indicates that $E_g$-symmetric strain mixes orbitals within the GS and ES manifolds, and thus phonon modes with corresponding strain components can induce resonant transitions between these orbitals. In contrast, strain along the SiV axis ($A_{1g}$-symmetric strain) is found to leave the GS and ES splittings unchanged, and therefore cannot cause electronic transitions.

**Strain-tuning of energy levels.** With our device we can tune the splitting of the orbitals in the GS manifold from 46 GHz to typically up to 500 GHz, and in the best case, up to 1.2 THz (see Supplementary Discussion). In doing so, we can probe the interaction between the colour centre and phonons of different frequencies. This is achieved by measuring the thermal relaxation rate between the orbitals with a time-resolved pump-probe technique (Fig. 2b). Measurements are performed in the frequency range $\Delta_{gs} = 46$ to 110 GHz where this technique can be applied. The total relaxation rate is a sum of the rates of phonon absorption, $\gamma_{up}$, and emission, $\gamma_{down}$ (shown in Fig. 1a), which can be individually extracted using the theory described in Supplementary Discussion. Over the range of $\Delta_{gs}$ measured, phonon processes in both directions are observed to accelerate with increasing orbital splitting. This is because the number of acoustic modes resonant with the GS splitting, i.e. the phonon density of states (DOS) at $\Delta_{gs}$, increases with the dependence $\Delta_{gs}^n$ ($n$ depends on the geometry of material seen by resonant phonons, see Supplementary Discussion). However, if the orbital splitting is increased far above 120 GHz (at temperature $T = 4$ K) as plotted in Fig. 2c, the phonon absorption rate ($\gamma_{up}$) is theoretically expected to reverse its initial trend. In this regime, the polynomial increase in phonon DOS is outweighed by the exponential decrease in thermal phonon occupation ($\sim \exp(-h\Delta_{gs}/k_B T)$)[11], and consequently $\gamma_{up}$ is rapidly quenched.

**Strain-enhanced spin coherence.** Such a suppression of phonon absorption at high strain can improve the spin coherence of the emitter. In the presence of a magnetic field, the SiV⁻ electronic levels further split into spin sub-levels and provide an optically accessible spin qubit as shown in Fig. 3a[23,30–32]. We use coherent population trapping (CPT) through simultaneous resonant laser excitation of the optical transitions labeled C1 and C2 to pump the SiV⁻ into a dark state, a coherent superposition of the spin sub-levels $|1\downarrow\rangle$, $|1\uparrow\rangle$[31,32]. When the two-photon detuning is scanned, preparation of the dark state results in a fluorescence dip, whose linewidth is determined by the optical driving and spin dephasing rates. At low laser powers, the linewidth is limited by spin dephasing, which is dominated by phonon-mediated transitions within the GS manifold[11,23]. In Fig. 3b, as the dark resonance narrows down due to prolonged spin coherence with increasing strain, we reveal a fine structure not visible before. Further measurements in Supplementary Discussion suggest that the presence of two resonances is due to the interaction of the SiV

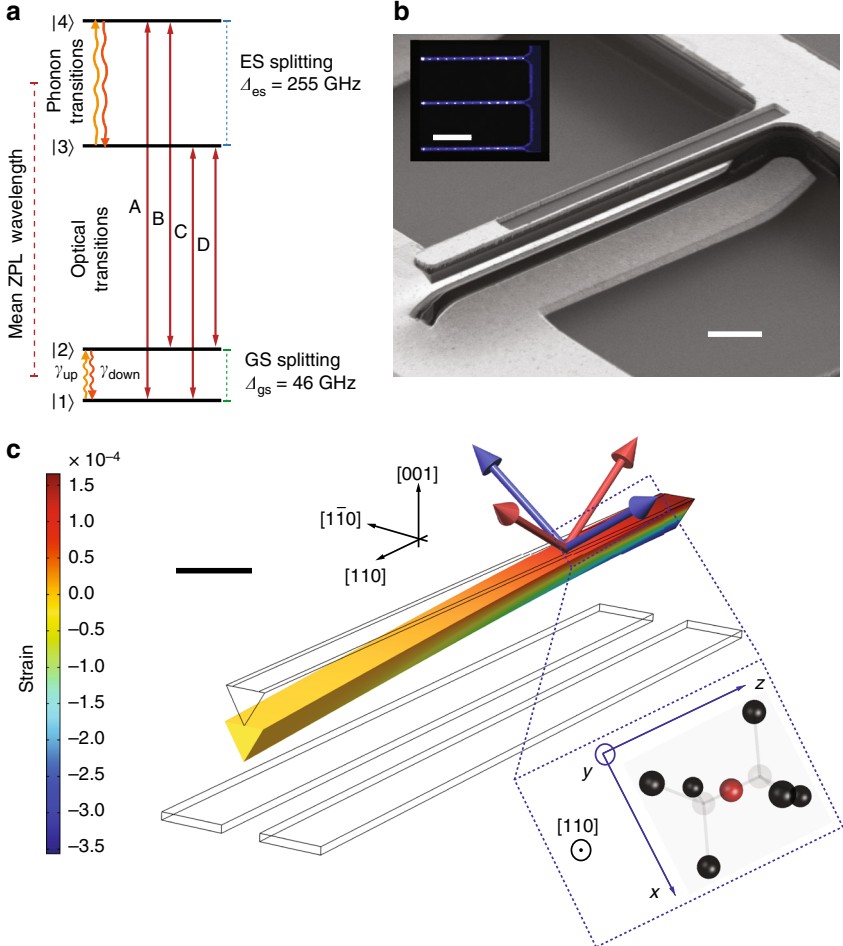

**Fig. 1** Description of the SiV⁻ and NEMS system. **a** Electronic level structure of the SiV⁻ showing the mean zero phonon line (ZPL) wavelength, frequency splittings between orbital branches in the ground state (GS) and excited state (ES) ($\Delta_{gs}$ and $\Delta_{es}$, respectively) at zero strain, and the four optical transitions A, B, C, and D. Also shown are single-phonon transitions in the GS and ES manifolds. **b** Scanning electron microscope (SEM) image of a representative diamond NEMS cantilever. Dark regions correspond to diamond, and light regions correspond to metal electrodes. Scale bar corresponds to 2 μm. (Inset) Confocal photoluminescence image of three adjacent cantilevers. The array of bright spots in each cantilever is fluorescence from SiV⁻ centres. Inset scale bar corresponds to 10 μm. **c** Simulation of the displacement of the cantilever due to the application of a DC voltage of 200 V between the top and bottom electrodes. The component of the strain tensor along the long axis of the cantilever is displayed using the colour scale. Scale bar corresponds to 2 μm. Crystal axes of diamond are indicated in relation to the geometry of the cantilever. Arrows on top of the cantilever indicate the highest symmetry axes of four possible SiV⁻ orientations, and their colour indicates separation into two distinct classes upon application of strain. SiV⁻ centres studied in this work are shown by blue arrows and are oriented along [1̄11], [1̄1̄1] directions. They are orthogonal to the cantilever long-axis, and experience strain predominantly in the plane normal to their highest symmetry axis. Inset shows the molecular structure of such a transverse-orientation SiV⁻ along with its internal axes, when viewed in the plane normal to the [110] axis

⁻ electron spin with a neighbouring spin such as a $^{13}$C nuclear spin. This indicates the possibility of achieving a local register of qubits as has been demonstrated with NV centres[33]. Figure 3c shows the decreasing linewidths of the CPT resonances with increasing GS orbital splitting, indicating an improved spin coherence time. Beyond a GS splitting of ~400 GHz, the linewidths saturate at ~1 MHz. At the highest strain condition, we perform a power-dependent CPT measurement to eliminate the contribution of power broadening, and extract a spin coherence time of $T_2^* = 0.25 \pm 0.02$ μs (compared with other CPT-based measurements which reported $T_2^* = 40$ ns without strain control[31,32]). This saturation of $T_2^*$ suggests the mitigation of the primary dephasing source, single-phonon transitions between the GS orbitals, and the emergence of a secondary dephasing mechanism such as slowly varying magnetic fields from naturally abundant (1.1%) $^{13}$C nuclear spins in diamond. We note that our longest $T_2^* = 0.25 \pm 0.02$ μs is on par with that of the NV⁻ centre without dynamical decoupling[4,34] and of low-strain SiV⁻ centres

operated at a much lower temperature of 100 mK[16], the conventional approach to suppress phonon-mediated dephasing.

## Discussion

In conclusion, we use a nano-electro-mechanical system to probe and control the interaction between a single electronic spin and the phonon bath of its solid-state environment. In doing so, we demonstrate a six-fold prolongation of spin coherence by suppressing phonon-mediated dephasing as the dominant decoherence mechanism. As a next step, we can further improve the spin coherence by cancelling the effect of slowly varying non-Markovian noise from the environment[16] using dynamical decoupling techniques that are well-studied with other spin systems[6,7,33]. Our strain engineering approach can be applied to overcome phonon-induced decoherence in other emitters such as emerging inversion-symmetric centres in diamond[13,14,35,36] Kramers rare earth ions[15,18,19], and in general, systems with spin-

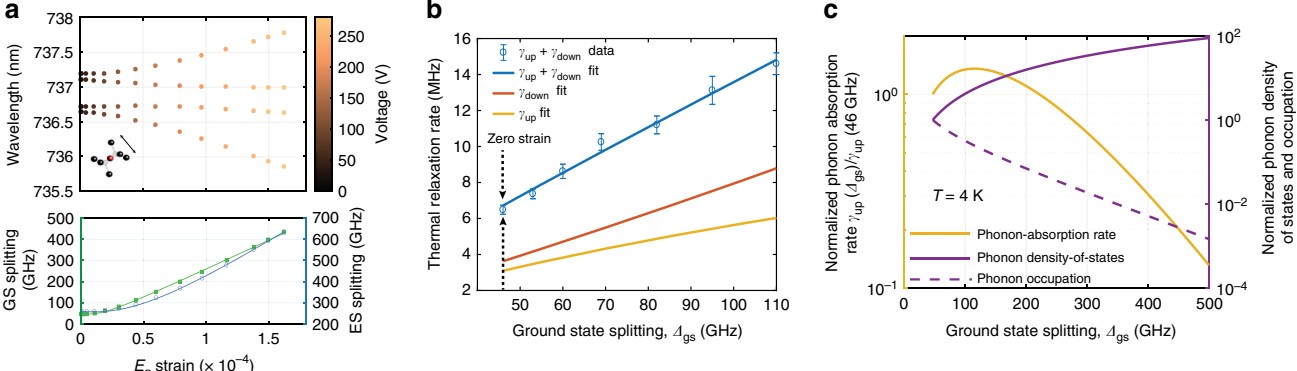

**Fig. 2** Strain-tuning of the SiV⁻ energy levels. **a** Strain response of a transverse-orientation SiV⁻ as shown in Fig. 1c. Wavelengths of the four optical transitions A, B, C, and D are recorded against strain. Raw PLE data with applied voltages can be found in Supplementary Fig. 7. The lower panel shows orbital splittings within GS (solid green squares) and ES (open blue circles) extracted from the optical transition wavelengths. Solid curves are fits to group theory-based strain response model.[27, 29] **b** Thermal relaxation rates between GS orbital branches vs. their energy splitting. Error bars represent standard deviation of the estimated rate, and are under 5% for all data points. Fit to model in Supplementary Discussion allows extraction of the phonon-absorption rate $\gamma_{up}$ and phonon-emission rate $\gamma_{down}$. **c** Calculated phonon-absorption rate $\gamma_{up}(\Delta_{gs})$ (solid yellow line) as a function of GS-orbital splitting $\Delta_{gs}$ at temperature $T = 4$ K. Left $y$-axis indicates the magnitude of this rate normalized to the value at zero strain, $\gamma_{up}(46$ GHz$)$. Right $y$-axis indicates the two competing factors whose product determines $\gamma_{up}$: the phonon density of states (normalized to its value at zero strain), shown with the solid violet line, and the thermal occupation of acoustic modes shown with the dashed violet line

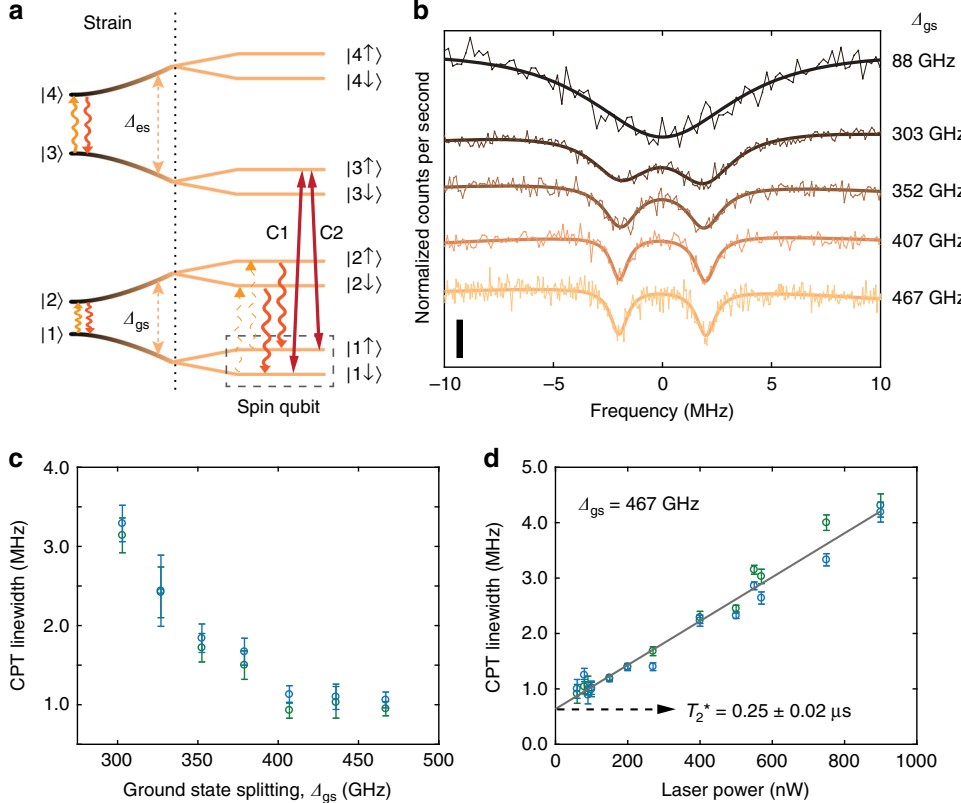

**Fig. 3** Spin coherence measurements. **a** SiV⁻ level structure in the presence of strain and external magnetic field. A spin qubit is defined with levels $|1\downarrow\rangle$ and $|1\uparrow\rangle$ on the lower orbital branch of the GS. This qubit can be polarized, and prepared optically using the Λ-scheme provided by transitions C1 and C2. Phonon transitions within ground- and excited-state manifolds are also indicated. The upward phonon transition (phonon absorption process) can be suppressed at high strain, thereby mitigating the effect of phonons on the coherence of the spin qubit. **b** Coherent population trapping (CPT) spectra probing the spin transition at increasing values of the GS orbital splitting $\Delta_{gs}$ from top to bottom. Scale bar in bottom left represents a fluorescence signal contrast of 10%. Measurements are carried till the noise in the fluorescence signal is below 1.5%. Bold solid curves are Lorentzian fits. Optical power is adjusted in each measurement to minimize power-broadening. **c** Linewidth of CPT dips as a function of GS orbital splitting $\Delta_{gs}$ indicating improvement in spin coherence with increasing strain. Error bars represent standard deviation of the estimated linewidths from the Lorentzian fits. **d** Power dependence of CPT-linewidth at the highest strain condition ($\Delta_{gs} = 467$ GHz). Data points are estimated linewidths from CPT measurements, and the solid curve is a linear fit, which reveals a linewidth of $0.64 \pm 0.06$ MHz corresponding to $T_2^* = 0.25 \pm 0.02$ μs

orbit coupling in their ground state. The high strain needed to quench phonon processes can be achieved simply by deposition of a thin film[37], which passively stresses the underlying crystal. A NEMS platform such as the one demonstrated in this paper can provide the added benefit of active wavelength tuning, which can enable generation of indistinguishable photons from multiple emitters, and hence scalable photonic quantum networks[20,38]. Another natural extension of our work is coherent coupling of the SiV⁻ spin to phonons in a well-defined mechanical mode, which will enable the use of phonons as a quantum resource. In particular, we can combine the large strain susceptibility of the SiV⁻ electronic levels with mechanical resonators of dimensions close to the phonon wavelength, such as optomechanical crystals[39] to obtain orders of magnitude larger spin-phonon interaction strengths compared with previous works[40–45], leading to strong spin-phonon coupling[29]. In this regime, one can realise phonon-mediated two-qubit gates[46,47] analogous to those implemented with trapped ions[48], and achieve quantum non-linearities required to deterministically generate single phonons and non-classical mechanical states[49–53], a long sought-after goal since phonons can be used to interface spins with other quantum systems such as superconducting qubits[54].

## Methods

**Fabrication procedure.** We use ⟨100⟩-cut, ultra-high purity (nitrogen concentration less than 5 ppb), type IIa, single-crystal diamond synthesized by chemical vapour deposition (CVD) from Element Six Corporation. The cantilever arrays are patterned with electron-beam lithography, and first vertically etched with oxygen plasma. These vertically etched structures are then made free-standing by etching the sample at a tilted angle with an oxygen-plasma assisted ion-milling process. After cantilever fabrication, silicon ions (Si⁺) are implanted at target spots on the cantilevers using a custom focused-ion-beam (FIB) system at Sandia National Labs. SiV centres are then generated by a high-temperature (1100 °C), high-vacuum annealing procedure followed by a tri-acid clean (1:1:1 sulfuric, perchloric, and nitric acids). Subsequently, electrode patterns are made by a conventional bi-layer PMMA process followed by metal evaporation. We use as a 10 nm thick tantalum (Ta) layer for the cantilever electrodes in order to reliably apply high voltages, and a 200 nm thick gold layer for the bonding pads. Detailed schematics for the above fabrication steps are shown in Supplementary Methods.

**Strain dependent photoluminescence measurements.** The sample is cooled down to a nominal temperature of 6 K inside a Janis ST-500 continuous helium-flow cryostat. The cryostat is mounted under a home-built scanning confocal microscope with a 0.9 NA ×100, 1 mm working distance objective (Olympus MPLFLN 100X) housed inside the cryostat. SiV centers are identified via non-resonant excitation with a 703 nm laser diode (Thorlabs LP705-SF15), and collection of zero-phonon-line (ZPL) fluorescence in a narrow bandwidth of 10 nm around 737 nm. For resonant photoluminescence excitation (PLE) of ZPL transitions, we use a tunable continuous-wave Ti-sapphire laser (M-Squared Solstis), and collect the resulting fluorescence in the phonon-sideband (above 750 nm). Mode-hop-free tuning of the laser is achieved over the scan range of interest by using feedback from a wavemeter (High Finesse WS7). A CW 532 nm laser is periodically pulsed using an acousto-optic modulator (Crystal Technology 3080) to maintain the negative charge state of SiV(−) centers. Collected fluorescence (gated off during green excitation) is sent to an avalanche-photodiode to measure the photon-count rate. DC voltage for cantilever-deflection is supplied from a Stanford Research Systems PS300 high-voltage source. As an added precautionary measure, the weak leakage-current in the circuit (typically below 100 nA) discussed in Supplementary Methods is monitored via a Keithley 2400 source-meter.

**Orbital thermalization measurements.** The pump-probe pulse sequence to measure the orbital thermalization rate is implemented by pulsing our resonant-excitation laser with a Mach-Zehnder intensity electro-optic modulator (EOM) (EO Space AZ-AV5-5-PFA-PFA-737). The EOM is driven by a digital-delay generator with rise- and fall-times of 2 ns (SRS DG645). Over the course of the measurements, the modulation index of the EOM is stabilized against long-term drifts with continuous feedback on the DC-bias voltage. The feedback loop is implemented with a lock-in amplifier (SRS SR830) generating a low-frequency (1 KHz) modulation of the DC-bias voltage. Photon-count pulses from the single-photon-detector are time-tagged on a PicoHarp 300 module triggered by the delay-generator. The laser frequency itself is stabilized by continuous feedback with a wavemeter (High Finesse WS7).

**SiV spin measurements.** The sample is cooled down to a temperature of 3.8 K inside a closed-cycle liquid helium cryostat (Attodry 1000). It is placed in a dip stick, in which helium gas (pressure ~1 mbar) acts as an exchange gas. Two superconducting coils surrounding the sample chamber can be used to apply a magnetic field along two orthogonal axes, up to 8 T vertically and up to 2 T horizontally. The optical part of the setup consists of a home-built confocal microscope mounted on top of the cryostat, and a microscope objective (NA = 0.82) inside the sample chamber. The sample is mounted on piezoelectric stages (Attocube ANPx101 and ANPz101) allowing to position the sample with respect to the objective. Non-resonant excitation of SiVs is performed using a diode laser at 660 nm (Laser Quantum Ventus), while resonant excitation is achieved with a tunable diode laser around 737 nm (Toptica DLpro). The frequency of the latter is stabilised through continuous feedback from a wavemeter (High Finesse WSU). For CPT measurements, sidebands are generated on the resonant excitation laser using an EOM (Photline NIR-MX800) connected to a tunable microwave source (Rhode&Schwarz SMF 100A). Fluorescence from the emitters is collected through the microscope objective. A 750 nm long-pass filter in the confocal microscope allows collection of the phonon-sideband emission from SiV⁻ centres, filtering out the laser excitation. This emission is then sent to an avalanche photodiode (APD) (Excellitas).

**Data availability.** The datasets generated during and/or analysed during the current study are available from the corresponding author on reasonable request.

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

## Acknowledgements

This work was supported by STC Center for Integrated Quantum Materials (NSF Grant No. DMR-1231319), ONR MURI on Quantum Optomechanics (Award No. N00014-15-1-2761), NSF EFRI ACQUIRE (Award No. 5710004174), the University of Cambridge, the ERC Consolidator Grant PHOENICS, and the EPSRC Quantum Technology Hub NQIT (EP/M013243/1). B.P. thanks Wolfson College (University of Cambridge) for support through a research fellowship. Device fabrication was performed in part at the Center for Nanoscale Systems (CNS), a member of the National Nanotechnology Infra-structure Network (NNIN), which is supported by the National Science Foundation under NSF award no. ECS-0335765. CNS is part of Harvard University. Focused ion beam implantation was performed under the Laboratory Directed Research and Development Program and the Center for Integrated Nanotechnologies, an Office of Science (SC) user facility at Sandia National Laboratories operated for the DOE (contract DE-NA0003525) by National Technology and Engineering Solutions of Sandia, LLC, a wholly owned subsidiary of Honeywell International, Inc. We thank D. Perry for performing the focused ion beam implantation, and K. De Greve and M. W. Doherty for helpful discussions.

## Author contributions

S.M., M.A. and M.L. conceived the experiment. M.A. and M.L. supervised the project. Y.-I.S.designed and modelled the devices. Y.-I.S. did electron-beam lithography of samples; H.A.A. did the angled etching; J.L.P., J.A. and E.B. did the targeted ion implantation; S.M. did the annealing; and Y.I.S. did the metal electrode fabrication. S.M.built the experimental setup for strain-dependent spectroscopy with assistance from Y.-I.S., J.H. and M.Z. B.P. built the experimental setup for spin measurements with assistance from M.G. and C.S. Y.-I.S. and S.M. performed strain-dependent spectroscopy and orbital thermalization measurements. Y.-I.S., S.M. and B.P. performed CPT measurements with assistance from M.G., C.S. and M.J.S. Y.-I.S. and S.M. analyzed strain-dependent photoluminescence and orbital thermalization data. Y.-I.S., S.M. and B.P. analysed the CPT data. A.S., J.C., M.D.L., M.A. and M.L. participated in discussion of results. Y.-I.S., S.M. and B.P. wrote the manuscript in discussion with all authors.

## Additional information

**Competing interests:** The authors declare no competing interests.

