## [Peer Review File · Nature Communications]

Reviewers' comments:

Reviewer #1 (Remarks to the Author):

What is the effect of Strain along the the orthogonal axes during the bending of the cantilever. Is there a calculation, simulation or estimation of Strain_XX and Strain_YY. It is assumed that the strain is only along one direction? Can you please provide simulation, measurement or both to justify this?

Reviewer #2 (Remarks to the Author):

The paper "Controlling the coherence of a diamond spin qubit through its strain environment" by Sohn et al. proves that by controlling the electron-phonon interaction using static strain thus suppressing phonon-mediated dephasing as the dominant decoherence mechanism, which is normally reduced by cryogenic operation in solid state systems. This is achieved with excellent control of the system integrated within a diamond NEMS. This control can be used also to tune the emission of single defects and can permit to establish indistinguishable single photons from different colour centres. The final spin decoherence $T_{2\text{star}}$ achieved is of 0.25 microsec at 4K. I found that the paper is very important and is an excellent contribution to provide a better understanding of the spin physics of the SiV in diamond. However the experiment is still performed at 4K if I did not miss something.

How does this specific colour centres compare to the NV centre in diamond and where are the advantages in implementing such a complicated experiment/fabrication if the low temperature operation is still needed?

Is the electron-photon interaction controllable by strain in this specific colour centre due to the inversion symmetry or can this be applied to other colour centres? It would be useful to know if this approach could be applied to other emerging or well established colour centres or at least speculate on the universality of the method.

I think the paper should address these major questions of general interest somewhere to be accepted in Nature Communication, otherwise I see it for a more specialised physics journal.

Reviewer #3 (Remarks to the Author):

The manuscript by Sohn et al. describes experiments on an isolated, negatively-charged silicon-vacancy center (SiV-) in diamond. In this work, they combine their existing expertise in fabricating diamond nanobeams with electrode deposition to enable strain control by applying an electric field; the coherence lifetime ($T_{2\text{*}}$) of the emitter is then investigated as a function of applied strain and found to be increased to 250 ns, close to the ~ 300 ns achieved at 100 mK in experiments on SiV- embedded in natural abundance ^{13}C material [1].

This work is an experimental realization of a strain control scheme described in an earlier Nat. Commun. paper by some of the present authors [2]. This work demonstrates a proof-of-principle in extending SiV- lifetimes through phonon suppression, and demonstrates electronic tuneability of single center emission frequency – the latter development could be critical for the realization of quantum networks of SiV- centers given its sensitivity to strain. The present work is timely and of interest to the diamond photonic and quantum communication communities; I therefore recommend it for publication in Nature Communications.

I have the following suggestions / questions for the authors:

- Line 73: What is the limit to the pump-probe technique? The wording suggests that 110 GHz is a

fundamental upper limit to this approach.

- At several locations in the manuscript, references are made to a future paper (Meesala et al.): it appears this paper is not yet on arXiv. The references to the unpublished paper are not critical to understand the conclusions or primary outcomes of the present work, and therefore I suggest that they ought to be removed. In particular, the inclusion in the SI of raw PLE data with applied voltages (used to generate the top panel of Fig. 2a) and the removal of the lower panel of Fig. 2a would strengthen the current work.
- Ref. [24] should be updated from arXiv to the final PRL reference [1]

[1] D. D. Sukachev et al., Phys. Rev. Lett. 119, 223602 (2017)

[2] B. Pingault et al., Nat. Commun. 8, 15579 (2017)

Our responses to the reviews are indicated in blue below.

Reviewer #1 (Remarks to the Author):

What is the effect of Strain along the the orthogonal axes during the bending of the cantilever. Is there a calculation, simulation or estimation of Strain_XX and Strain_YY. It is assumed that the strain is only along one direction? Can you please provide simulation, measurement or both to justify this?

We thank the reviewer for this important question. We have indeed carried out more extensive measurements and simulations to characterize the complete strain response of the SiV center considering the effects of all strain tensor components. Discussion of the full strain response requires group theory analysis, measurements, simulations, and data fitting for both types of emitter orientations – ‘axial’ and ‘transverse’, as shown by red and blue arrows in Fig. 1c. Since the focus of the present manuscript is control of phonon-induced decoherence, these details have been reserved for a separate paper, which will be submitted to a more specialized physics journal. An arXiv pre-print of this paper has been attached for the reviewer’s consideration. This has been cited as Ref. [37] in line 71 of the revised manuscript for readers interested in the full details of the strain response.

The reviewer is directed to Sections II and III of this arXiv pre-print, which describe our procedure to deduce the strain Hamiltonian. Specifically, Fig. 3(c) shows the simulated strain tensor components for the ‘transverse’ orientation SiV studied in the current manuscript. Strain components orthogonal to the cantilever axis are a small fraction (~ 0.2 corresponding to the Poisson ratio) of the component along the axis. While these small terms are included in our group theory based model and fitting procedure, our final results reveal that the physics is qualitatively well-described by just the one strain component along the cantilever axis. This component alone determines $\sim 96\%$ of E_g symmetric strain for transverse orientation SiVs, hence controls the orbital splitting in the ground state of the emitter, and consequently impacts phonon-induced decoherence. The other mode of strain (A_{1g} strain) was found to not influence the orbital splittings, and therefore phonons of this type cannot cause electronic transitions.

In the revised manuscript, we explicitly state this observation (see lines 68, 69) to suggest that these strain components are inessential to control phonon-induced decoherence. We hope this added text along with the updated citation makes the discussion on cantilever-induced strain more technically sound, while maintaining the elegant picture of only one strain component (of all six) being physically significant.

Reviewer #2 (Remarks to the Author):

The paper "Controlling the coherence of a diamond spin qubit through its strain environment" by Sohn et al. proves that by controlling the electron-phonon interaction using static strain thus suppressing phonon-mediated dephasing as the dominant decoherence mechanism, which is normally reduced by cryogenic operation in solid state systems. This is achieved with excellent control of the system integrated within a diamond NEMS. This control can be used also to tune the emission of single defects and can permit to establish indistinguishable single photons from different colour centres. The final spin decoherence T2star achieved is of 0.25 microsec at 4K. I found that the paper is very important and is an excellent contribution to provide a better understanding of the spin physics of the SiV in diamond. However the experiment is still performed at 4K if I did not miss something. How does this specific colour centres compare to the NV centre in diamond and where are the advantages in implementing such a complicated experiment/fabrication if the low temperature operation is still needed?

Is the electron-photon interaction controllable by strain in this specific colour centre due to the inversion symmetry or can this be applied to other colour centres? It would be useful to know if this approach could be applied to other emerging or well established colour centres or at least speculate on the universality of the method.

I think the paper should address these major questions of general interest somewhere to be accepted in Nature Communication, otherwise I see it for a more specialised physics journal.

We thank the reviewer for their very relevant comments. The operating temperature in our experiments is indeed 4K, which is on par with or above that of other solid-state quantum memories with optical access. This includes NV centers, III-V quantum dots, and rare earth ions. It is important

to note that key requirements for photonic quantum networks, such as remote entanglement, quantum teleportation, and spin-photon state transfer rely on generation of coherent optical photons from these emitters. To maintain good optical properties such as high zero-phonon-line fluorescence, and nearly radiatively broadened emission, cryogenic temperatures in the *few Kelvin to tens of Kelvin* range are required *regardless of the exact emitter chosen*. As a result, experiments geared towards quantum photonic networks are typically carried out in helium-4 cryostats. In the specific case of the well-known diamond NV center, these operating conditions are required to obtain coherent photons despite the NV center being a long-lived quantum memory and excellent magnetic field sensor at room temperature. This is why all demonstrations of stationary-flying qubit interfacing have been performed at cryogenic temperatures (Nature 497 (7447), 86 (2013), Science 345 (6196), 532-535 (2014), Nature 526 (7575), 682-686 (2015), Science 356 (6341), 928-932 (2017)).

Our argument is that sensitivity of the quantum memory to phonons can demand even lower, sub-Kelvin operating temperature to achieve long coherence and relaxation times. This is a stringent demand since it requires significantly more complex and expensive cryogenic setups such as dilution refrigerators instead of simpler helium-4 cryostats, which are the current workhorse of the field. The challenge is not restricted to newly emerging, optically superior inversion-symmetric centers such as the silicon-, germanium- and tin-vacancy defects in diamond, but also for more extensively studied rare-earth ion memories (eg. see Refs. [25,27,28] in the manuscript), and more generally, for systems with spin-orbit coupling in the ground state. Our strain engineering approach is applicable to all these platforms.

Moreover, now that controlled study with our NEMS device has shown that high strain can suppress such detrimental electron-phonon processes, future work on any of these platforms can use passive material strain to achieve the same goal. This can be achieved by coating the relevant sample with a high-stress thin film (eg. silicon nitride), a relatively straightforward process compared to the full-fledged fabrication of a NEMS device, which is only required for active strain control. Thus, in addition to mentioning the effect of phonon processes on experimental complexity in the introduction (lines 32-36), we have added sentences to the conclusion (see lines 117-122) to suggest the general applicability of strain engineering to other systems, and to describe a simple approach for the same.

As a final note, the interest in optically superior, inversion symmetric centers such as the SiV began largely due the poor optical properties of the NV center even at cryogenic temperatures (4 K). The advantages of such inversion symmetric centers include stronger emission into the zero-phonon line (about 80% of the total fluorescence for SiV compared to 4% for NV at 4K) and spectral stability owing to their insensitivity to electric field noise to first order. Relevant papers that expand on these optical properties have been cited in the current manuscript, eg. Refs. [29, 30]. An overarching goal of these investigations is to demonstrate an emitter that simultaneously provides excellent optical properties and a long-lived spin qubit. Our work is a crucial step in this direction, since we show good spin coherence with an optically well-behaved emitter. The spin coherence time (T_2^* of 0.25 microseconds) we demonstrate after strain engineering is comparable to that of NV centers without the use of dynamical decoupling techniques. In the revised manuscript, we have explicitly stated this benchmark in lines 109-110.

Reviewer #3 (Remarks to the Author):

The manuscript by Sohn et al. describes experiments on an isolated, negatively-charged silicon-vacancy center (SiV-) in diamond. In this work, they combine their existing expertise in fabricating diamond nanobeams with electrode deposition to enable strain control by applying an electric field; the coherence lifetime (T_2^*) of the emitter is then investigated as a function of applied strain and found to be increased to 250 ns, close to the ~300 ns achieved at 100 mK in experiments on SiV- embedded in natural abundance ^{13}C material [1].

This work is an experimental realization of a strain control scheme described in an earlier Nat. Commun. paper by some of the present authors [2]. This work demonstrates a proof-of-principle in extending SiV- lifetimes through phonon suppression, and demonstrates electronic tuneability of single center emission frequency – the latter development could be critical for the realization of quantum networks of SiV- centers given its sensitivity to strain. The present work is timely and of interest to the diamond photonic and quantum communication communities; I therefore recommend it

for publication in Nature Communications.

We thank the reviewer for their encouraging comments, and address their questions below.

I have the following suggestions / questions for the authors:

- Line 73: What is the limit to the pump-probe technique? The wording suggests that 110 GHz is a fundamental upper limit to this approach.

The pump-probe technique mentioned by the reviewer is used to measure the thermalization rate between the ground state orbital branches 1 and 2 (as shown in Fig. 1a). The measurement begins with a pump pulse resonant with optical transition D. The goal here is to deplete the initial thermal population in level 2, and initialise the SiV entirely in level 1. As the ground state splitting is increased above its zero strain value (46 GHz), the thermal rates (γ_{down} and γ_{up} in the manuscript) increase (see Figs. 2b and 2c) with increasing phonon density of states. Eventually, the rate γ_{down} approaches the excited state optical decay rate, and the ground state population cannot be optically pumped to a condition better than its nominal thermal distribution. The probe fluorescence signal corresponds to a recovering population difference between the two ground state branches with the thermal distribution corresponding to the steady state after the pump pulse. As the efficiency of optical pumping reduces, it becomes increasingly difficult to distinguish between the optically pumped case and the thermal steady state, and therefore to measure the thermalization rate. To clarify this point, we have included detailed information on the pump-probe measurements in Section 5 of the SI. Specifically, Fig. S5 shows the inability of optical pumping to outpace the thermal decay at high strain.

- At several locations in the manuscript, references are made to a future paper (Meesala et al.): it appears this paper is not yet on arXiv. The references to the unpublished paper are not critical to understand the conclusions or primary outcomes of the present work, and therefore I suggest that they ought to be removed. In particular, the inclusion in the SI of raw PLE data with applied voltages (used to generate the top panel of Fig. 2a) and the removal of the lower panel of Fig. 2a would strengthen the current work.

The paper is now available on arXiv, and the relevant citation Ref. [37] in the revised manuscript has been updated. We have attached a copy of the same for the reviewer's consideration. We feel that this reference i.e. Ref. [37] in the revised manuscript should be retained, since it discusses the full SiV strain response, and strain-mediated couplings between various energy levels fundamentally responsible for phonon decoherence. These details exactly answer the questions raised by another reviewer (reviewer 1).

The lower panel of Fig. 2a emphasizes the ability to increase ground state splitting with strain. This is the key requirement to control phonon transitions, and improve spin coherence – all results that appear in subsequent panels of Fig. 2 and Fig. 3. It is tricky for the uninitiated reader to immediately deduce increasing ground state splitting from the upper panel of Fig. 2a, so the lower panel was included for the sake of clarity. We do agree with the reviewer's suggestion that raw PLE data can add to clarity, and have included this in Section 3 in the revised SI.

- Ref. [24] should be updated from arXiv to the final PRL reference [1]

The reference has been updated in the revised version.

REVIEWERS' COMMENTS:

Reviewer #2 (Remarks to the Author):

I am satisfied with the revision and response to referees, and recommend publication in the present form in Nature Communications

Reviewer #3 (Remarks to the Author):

I am satisfied with the responses given to my queries. The additional information added to the SI and in relation to reference [37] strengthens the paper significantly. I recommend this paper be published in Nature Communications.